# List Approximation for Increasing Kolmogorov Complexity

Marius Zimand 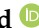

Department of Computer and Information Sciences, Towson University, Baltimore, MD 21252, USA; mzimand@towson.edu

**Abstract:** It is impossible to effectively modify a string in order to increase its Kolmogorov complexity. However, is it possible to construct a few strings, no longer than the input string, so that most of them have larger complexity? We show that the answer is yes. We present an algorithm that takes as input a string $x$ of length $n$ and returns a list with $O(n^2)$ strings, all of length $n$, such that 99% of them are more complex than $x$, provided the complexity of $x$ is less than $n - \log \log n - O(1)$. We also present an algorithm that obtains a list of quasi-polynomial size in which each element can be produced in polynomial time.

**Keywords:** Kolmogorov complexity; random strings; extractors

## 1. Introduction

The Kolmogorov complexity of a binary string $x$, denoted $C(x)$, is the minimal description length of $x$, i.e., it is the length of the shortest program (in a fixed universal programming system) that prints $x$. We analyze the possibility of modifying a string in an effective way in order to obtain a string with higher complexity, without increasing its length. Strings with high complexity exhibit good randomness properties and are potentially useful, because they can be employed in lieu of random bits in probabilistic algorithms. It is common to define the randomness deficiency of $x$ as the difference $|x| - C(x)$ (where $|x|$ is the length of $x$) and to say that the smaller the randomness deficiency is, the more random the string is. In this sense, we want to modify a string so that it becomes "more" random. As stated, the above task is impossible, because, clearly, any effective modification cannot increase the Kolmogorov complexity (at least not by more than a constant). If $f$ is a computable function, $C(f(x)) \leq C(x) + O(1)$, for every $x$. Consequently, we have to settle for a weaker solution and the one we consider is that of list approximation. List approximation consists in the construction of a list of objects guaranteed to contain at least one element having the desired property. Here, we try to obtain a stronger type of list approximation, in which, not just one, but *most* of the elements in the list have the desired property. More precisely, we study the following question.

*Question.* Is there a computable function which takes as input a string $x$ and outputs a short list of strings, which are not longer than $x$, such that most of the elements in the list have complexity greater than $C(x)$?

The formulation of the question rules out some trivial and non-interesting answers. First, the requirement that the list is "short" is necessary, because, otherwise, we can ignore the input $x$ and simply take all strings of length $n$ and most of them have complexity at least $n - 2$, which is within $O(1)$ of the largest complexity of strings of length $n$. Secondly, the restriction that the length is not increased is also necessary, because, otherwise, we can append to the input $x$ a random string and obtain, with high probability, a more complex string (see the discussion in Section 2). These restrictions not only make the problem interesting, but also amenable to applications in which the input string and the modified

strings need to be in a given finite set. The solution that we give can be readily adjusted to handle such applications.

There are several parameters to consider. The first one is the size of the list. The shorter the list is, the better the approximation is. Next, the increasing-complexity procedure that we seek does not work for all strings $x$. Let us recall that $C(x) \leq |x| + O(1)$ and, if $x$ is a string of maximal complexity at its length, then there simply is no string of larger complexity at its length. In general, for strings $x$ that have complexity close to $|x|$, it is difficult to increase their complexity. Thus, a second parameter is the bound on the complexity of $x$ for which the increasing-complexity procedure succeeds. The closer this bound is to $|x|$, the better the procedure is. The third parameter is the complexity of the procedure. The procedure is required to be computable, but it is preferable if it is computable in polynomial time.

We show the following two results. The first one exhibits a computable list approximation for increasing the Kolmogorov complexity that works for any $x$ with complexity $C(x) < |x| - \log\log|x| - O(1)$.

**Theorem 1** (Computable list of quadratic size for increasing the Kolmogorov complexity).
*There exists a computable function $f$ that takes as input $x \in \{0,1\}^*$ and a rational number $\delta > 0$ and returns a list of strings of length at most $|x|$ with the following properties:*

1.  *The size of the list is $O(|x|^2)\mathrm{poly}(1/\delta)$;*
2.  *If $C(x) < |x| - \log\log|x| - O(1)$, then the $(1 - \delta)$ fraction of the elements in the list $f(x)$ have a Kolmogorov complexity larger than $C(x)$ (where the constant hidden in $O(1)$ depends on $\delta$).*

Whether the bound $C(x) < |x| - \log\log|x| - O(1)$ can be improved remains open. Further reducing the list size is also an interesting open question. We could not establish a lower bound and, as far as we currently know, it is possible that even a constant list size may be achievable.

In the next result, the complexity-increasing procedure runs in polynomial time in the following sense. The size of the list is only quasi-polynomial, but each string in the list is computed in polynomial time.

**Theorem 2** (Polynomial-time computable list for increasing the Kolmogorov complexity).
*There exists a function $f$ that takes as input $x \in \{0,1\}^*$ and a constant rational number $\delta > 0$ and returns a list of strings of length at most $|x|$ with the following properties:*

1.  *The size of the list is bounded by $2^{O(\log|x| \cdot \log(|x|/\delta))}$;*
2.  *If $C(x) < |x| - O(\log|x| \cdot \log(|x|/\delta))$, then $(1 - \delta)$ fraction of the elements in the list $f(x)$ have a Kolmogorov complexity larger than $C(x)$;*
3.  *The function $f$ is computable in polynomial time in the following sense: there is a polynomial time algorithm that takes as input $x, i$ and computes the $i$-th element in the list $f(x)$.*

**Remark 1.** *A preliminary version of this paper has appeared in STACS 2017 [1]. In that version, it was claimed that the result in Theorem 1 holds for all strings $x$ with $C(x) < |x|$. The proof had a bug and we can only prove it for strings satisfying $C(x) < |x| - \log\log|x| - O(1)$. The proof of Theorem 2 given here is different from that in [1]. Theorem 2 has better parameters than its analog in the preliminary version.*

**Remark 2.** *Any procedure that constructs the approximation list can be converted into a probabilistic algorithm that does the same work and picks one random element from the list. The procedure in Theorem 2 can be converted into a polynomial-time probabilistic algorithm, which uses $O(\log|x| \cdot \log(|x|/\delta))$ random bits to pick which element from the list to construct (see item 3 in the statement).*

*Vice-versa, a probabilistic algorithm can be converted into a list-approximation algorithm in the obvious way, i.e., by constructing the list that has as elements the outputs of the algorithm for all choices of the random coins.*

*Thus, a list-approximation algorithm $A_1$, in which $(1 - \delta)$ elements in the list have the desired property, is equivalent to a probabilistic algorithm $A_2$ that succeeds with probability $1 - \delta$. The number of random bits used by $A_2$ is the logarithm in base two of the size of the list produced by $A_1$.*

### 1.1. Basic Concepts and Notation

We recall the standard setup for Kolmogorov complexity. We fix an universal Turing machine $U$. The universality of $U$ means that, for any Turing machine $M$, there exists a computable "translator" function $t$, such that, for all strings $p$, $M(p) = U(t(p))$ and $|t(p)| \leq |p| + O(1)$. For the polynomial-time constructions, we also require that $t$ is polynomial-time computable. If $U(p) = x$, we say that $p$ is a *program* (or *description*) for $x$. The Kolmogorov complexity of the string $x$ is $C(x) = \min\{|p| \mid p \text{ is a program for } x\}$. If $p$ is a program for $x$ and $|p| \leq C(x) + c$, we say that $p$ is a *c-short program* for $x$.

### 1.2. Related Works

The problem of increasing the Kolmogorov complexity has been studied before by Buhrman, Fortnow, Newman and Vereshchagin [2]. They show that there exists a polynomial-time computable $f$ that takes as input $x$ of length $n$ and returns a list of strings, all having length $n$, such that, if $C(x) < n$, then there exists $y$ in the list with $C(y) > C(x)$ (this is Theorem 14 in [2]). In the case of complexity conditioned by the string length, they show that it is even possible to compute in polynomial time a list of constant size. That is, $f(x)$ is a list with $O(1)$ strings of length $n$ and, if $C(x \mid n) < n$, then it contains a string $y$ with $C(y \mid n) > C(x \mid n)$ (this is Theorem 11 in [2]). Our results are incomparable with the results in [2]. On one hand, their results work for any input $x$ with complexity less than $|x|$, while, in Theorem 1, we only handle inputs with complexity at most $|x| - \log \log |x| - O(1)$ (and, in Theorem 2, the complexity of the input is required to be even lower). On the other hand, they only guarantee that one string in the output list has higher complexity than $x$, while we guarantee this property for most strings in the output list and this can be viewed as a probabilistic algorithm with few random bits as explained in Remark 2.

This paper is inspired by recent list-approximation results regarding another problem in the Kolmogorov complexity, namely, the construction of short programs (or descriptions) for strings. Using a Berry paradox argument, it is easy to see that it is impossible to effectively construct a shortest program for $x$ (or, even a, say, $n/2$-short program for $x$). Remarkably, Bauwens et al. [3] show that effective list approximation for short programs is possible. There is an algorithm that, for some constant $c$, takes as input $x$ and returns a list with $O(|x|^2)$ strings guaranteed to contain a $c$-short program for $x$. They also show a lower bound; The quadratic size of the list is minimal up to constant factors. Bauwens and Zimand [4] consider a more general type of optimal compressor that goes beyond the standard Kolmogorov complexity and, using another type of pseudo-random function called *conductor*, re-obtains the overhead of $O(\log^2 n)$. Theorem 2 directly uses results from the latter, namely, Theorem 3. Theorem 1 uses a novel construction, but some of the ideas are inspired from the papers mentioned above.

## 2. Technique and Proof Overview

We start by presenting an approach that probably comes to mind first. It does not work for inputs $x$ having a complexity very close to $|x|$, such as in Theorem 1 (for which we use a more complicated argument), but, combined with the results from [4], it yields Theorem 2.

Given that we want to modify a string $x$ so that it becomes more complex, which, in a sense, means more random, a simple idea is to just append a random string $z$ to $x$. Indeed, if we consider strings $z$ of length $c$, then $C(xz) > C(x) + c/2$, for most strings $z$, provided

that $c$ is large enough. Let us see why this is true. Let $k = C(x)$ and let $z$ be a string that satisfies the opposite inequality, that is,

$$C(xz) \leq C(x) + c/2, \tag{1}$$

Given a shortest program for $xz$ and a self-delimited representation of the integer $c$, which is $2 \log c$ bits long, we obtain a description of $x$ with at most $k + c/2 + 2 \log c$ bits. Note that, in this way, from different $z$'s satisfying (1), we obtain different programs for $x$ that are $(c/2 + 2 \log c)$-short. By a theorem of Chaitin [5] (also presented as Lemma 3.4.2 in [6]), for any $d$, the number of $d$-short programs for $x$ is bounded by $O(2^d)$. Thus, the number of strings $z$ satisfying (1) is bounded by $2^{c/2 + 2 \log c + O(1)}$. Since, for large $c$, $2^{c/2 + 2 \log c + O(1)}$ is much smaller than $2^c$, it follows that most strings $z$ of length $c$ satisfy the claimed inequality (the opposite of (1)). Therefore, we obtain the following lemma.

**Lemma 1.** *If we append to a string $x$, a string $z$ chosen at random in $\{0, 1\}^c$, then $C(xz) > C(x) + c/2$ with probability $1 - 2^{-(c/2 - 2 \log c - O(1))}$.*

The problem with appending a random $z$ to $x$ is that this operation not only increases the complexity (which is something we want) but also increases the length (which is something we do not want). The natural way to get around this problem is to first compress $x$ to close to the minimal description length using the probabilistic algorithms from [4] described in the Introduction and then append $z$. If we know $C(x)$, then the algorithms from [4] compress $x$ to length $C(x) + \Delta(n)$, where $n$ is the length of $x$ and $\Delta(n)$ (called the *overhead*) is $O(\log n)$ (or poly$(\log n)$ for the polynomial-time algorithm). After appending a random $z$ of length $c$, we obtain a string of length $C(x) + \Delta(n) + c$ and, for this to be $n$ (so that length is not increased), we need $C(x) \leq n - \Delta(n) - c$. This is the idea that we follow for Theorem 2, with an adjustment caused by the fact that we do not know $C(x)$ but only a bound of it.

However, in this way, we cannot obtain a procedure that works for all $x$ with $C(x) < n - \log \log n - O(1)$, as required in Theorem 1. Our proof for this theorem is based on a different construction. The centerpiece is a type of bipartite graph with a low congestion property. Once we have the graph (in which the two bipartitions are called the set of left nodes and the set of right nodes), we view $x$ as a left node and the list $f(x)$ consists of some of the nodes at distance 2 from $x$ in the graph. (A side remark: Buhrman et al. [2] also use graphs, namely, constant-degree expanders, and they obtain the lists also as the set of neighbors at some given distance.) In our graph, the left side is $L = \{0, 1\}^n$, the set of $n$-bit strings, the right side is $R = \{0, 1\}^m$, the set of $m$-bit strings, and each left node has degree $D$. The graphs also depend on three parameters, $\epsilon, \Delta$ and $t$, and, for our discussion, it is convenient to also use $\delta = \epsilon^{1/2}$ and $s = \delta \cdot \Delta$. The graphs that we need have two properties:

- For every subset $B$ of left nodes of size at most $2^t$, the $(1 - \delta)$ fraction of nodes in $B$ satisfies the low congestion condition which requires that the $(1 - \delta)$ fraction of their right neighbors have at most $s$ neighbors in $B$. (More formally, for all $B \subseteq L$ with $|B| \leq 2^t$, for all $x \in B$, except at most $\delta|B|$ elements, all neighbors $y$ of $x$, except at most $\delta D$, have $\deg_B(y) \leq s$, where $\deg_B(y)$ is the number of $y$'s neighbors that are in $B$. We say that such $x$ has the low-congestion property for $B$.)
- Each right node has at least $\Delta$ neighbors.

The graph with the above two properties is constructed using the probabilistic method in Lemma 2.

Let us now see how to use such a graph to increase the Kolmogorov complexity in the list-approximation sense. Let us suppose that we have a graph $G$ with the above properties for the parameters $n, \delta, \Delta, D, s$ and $t$.

**Claim 1.** *There is a procedure that takes as input a string x of length n with complexity $C(x) < t$ and produces a list with $D \cdot \Delta$ strings, all having length n, such that at least a fraction of $(1 - 2\delta)$ of the strings in the list has a complexity larger than $C(x)$.*

Indeed, let $x$ be a string of length $n$ with $C(x) = k < t$. Let us consider the set $B = \{x' \in \{0,1\}^n \mid C(x') \leq k\}$, which we view as a set of left nodes in $G$. Note that the size of $B$ is bounded by $2^t$. A node that does not have the low-congestion property for $B$ is said to be $\delta$-BAD($B$). By the first property of $G$, there are at most $\delta|B|$ elements in $B$ that are $\delta$-BAD($B$). It can be shown that $x$ is not $\delta$-BAD($B$). The reason is, essentially, that the strings that are $\delta$-BAD($B$) can be enumerated and they make up a small fraction of $B$; therefore, they can be described with less than $k$ bits. Now, to construct the list, we view $x$ as a left node in $G$ and we "go-right-then-go-left". This means that we first "go-right", i.e., we take all the $D$ neighbors of $x$ and, for each such neighbor $y$, we "go-left", i.e., we take $\Delta$ of the $y$'s neighbors and put them in the list. Since $x$ is not $\delta$-BAD($B$), $(1 - \delta)D$ of its neighbors have at most $s = \delta \cdot \Delta$ elements in $B$. Overall, less than $2\delta \cdot D \cdot \Delta$ of the strings in the list can be in $B$ and so at least a fraction of $(1 - 2\delta)$ of the strings in the list has complexity larger than $k = C(x)$. Our claim is proved.

### 3. Proof of Theorem 2

We use the following definition and results from [4].

**Definition 1.**

- *A compressor $\mathcal{C}$ is a probabilistic function that takes as input a rational number $\epsilon > 0$, a positive integer m and a string x and outputs (with probability 1) a string $\mathcal{C}(\epsilon, m, x)$ of length exactly m.*
- *$\Delta(\epsilon, m, n)$ is a function of $\epsilon$ and positive integers m and n, called* overhead.
- *A compressor $\mathcal{C}$ is $\Delta$-optimal for the Kolmogorov complexity, if there exists an algorithm $\mathcal{D}$ (called* decompressor*) such that, for every string x, every rational $\epsilon \geq 2^{-|x|}$ and every $m \geq C(x) + \Delta(\epsilon, m, |x|)$,*

$$Prob[\mathcal{D}(\mathcal{C}(\epsilon, m, x)) = x] \geq 1 - \epsilon.$$

In other words, if we are given a bound $m$ that is at least $C(x)$+overhead, then $\mathcal{C}$ compresses $x$ to a string of length $m$, from which $\mathcal{D}$ is able to reconstruct $x$ with high probability.

**Theorem 3** (Theorem 1.1 in [4]). *There exists a compressor $\mathcal{C}$ with overhead $\Delta(\epsilon, m, n) = O(\log m \cdot \log(n/\epsilon))$ that is $\Delta$-optimal for the Kolmogorov complexity. Furthermore, the compressor $\mathcal{C}$ takes as input $(\epsilon, m, x)$ and runs in polynomial time in $|x|$, using a random string of length $O(\log m \cdot \log(|x|/\epsilon))$.*

Note: Theorem 1.1 in [4] is more general, but we only need the above version.

**Proof of Theorem 2.** We follow the plan sketched in Section 2; we compress the input $x$ to a string $y$ with the optimal compressor from Theorem 3 and then append to $y$ a random string $z$ of constant length. We show that, with high probability, $yz$ has the desired properties; it has a complexity larger than $C(x)$ and it is not longer than $x$. We see below that this randomized algorithm uses $O(\log |x| \cdot \log |x|/\epsilon))$ random bits, which implies the desired list approximation via the observations in Remark 2.

Let the compressor $\mathcal{C}$ and the overhead $\Delta$ be the functions from Theorem 3. Let $\epsilon = \delta/2$. We fix $n$; let us consider a string $x$ of length $n$ such that $C(x) \leq n - 3\Delta(\epsilon, n, n)$. Note that $C(x) \leq n - O(\log n \cdot \log(n/\epsilon))$. Let $m = n - 2\Delta(\epsilon, n, n)$ and $y = \mathcal{C}(\epsilon, m, x)$ (note that $y$ is a random variable because $\mathcal{C}$ is a randomized function). For $n$ sufficiently large,

$$C(x) \leq n - 3\Delta(\epsilon, n, n) \leq m - \Delta(\epsilon, m, n).$$

Let $\mathcal{A}$ be the event by which the decompressor $\mathcal{D}$ reconstructs $x$ from $y$. By Theorem 3, $\mathcal{A}$ has probability $1 - \epsilon$.

We take $c$ a constant large enough such that Equations (2) and (3) below are satisfied. Conditioned by $\mathcal{A}$,

$$C(y) \geq C(x) - c \text{ (because } x \text{ is reconstructed from } y) \tag{2}$$

Let $c' = 2c$. We choose $c$ so that

$$2^{-(c'/2 - 2\log c' - O(1))} < \epsilon, \tag{3}$$

where the $O(1)$ term is the constant from Lemma 1.

We append to $y$ a string $z$ chosen at random in $\{0,1\}^{c'}$. By Lemma 1 and Equation (3), with probability $1 - \epsilon$, $C(yz) > C(y) + c'/2 = C(y) + c$. Now, we condition on $\mathcal{A}$ and we obtain that, with probability $1 - 2\epsilon$,

$$C(yz) > C(y) + c \geq C(x) - c + c = C(x).$$

We take $\delta = 2\epsilon$. Now, let us check the properties of the above algorithm. For every $n$-bit string $x$ with $C(x) \leq n - 3\Delta(\epsilon, n, n) = n - O(\log |x| \cdot \log |x|/\delta)$, the algorithm takes as input $x$ and $\delta$ and outputs, in polynomial time, the string $yz$ that, with probability $1 - \delta$, has a complexity larger than the complexity of $x$. The string $yz$ has length $m + c = n - 2\Delta(\epsilon, n, n) + c \leq n$. The whole randomized procedure uses $O(\log m \cdot \log(n/\epsilon)) = O(\log n \cdot \log(n/\delta))$ random bits for compression with $\mathcal{C}$ and $c' = O(1)$ random bits for $z$. The list approximation is obtained from the probabilistic algorithm in the obvious way, i.e., by including in the list one element for each choice of the random string (see Remark 2). The theorem is proved. □

## 4. Proof of Theorem 1

We split the proof in three parts. In Section 4.1, we introduce *balanced graphs*; in Section 4.2, we show how to increase the Kolmogorov complexity in the list approximation sense using balanced graphs and, in Section 4.3, we use the probabilistic method to obtain the balanced graph with the parameters needed for Theorem 1.

### 4.1. Balanced Graphs

Here, we formally define the type of graphs that we need. We work with families of bipartite graphs $G_n = (L \cup R, E \subseteq L \times R)$, indexed by $n$, which have the following structure:

1. The vertices are labeled with binary strings, $L = \{0,1\}^n$ and $R = \{0,1\}^n$, where we view $L$ as the set of left nodes and $R$ as the set of right nodes.
2. All the left nodes have the same degree $D$; $D = 2^d$ is a power of two and the edges outgoing from a left node $x$ are labeled with binary strings of length $d$.
3. We allow multiple edges between two nodes to exist. For a node $x$, we write $N(x)$ for the *multiset* of $x$'s neighbors, each element being taken with the multiplicity equal to the number of edges from $x$ landing into it.

A bipartite graph of this type can be viewed as a function $\text{EXT} : \{0,1\}^n \times \{0,1\}^d \rightarrow \{0,1\}^n$, where $\text{EXT}(x,y) = z$ if there is an edge between $x$ and $z$ labeled $y$. We want EXT to yield a $(k, \epsilon)$ randomness extractor whenever we consider the modified function $\text{EXT}_k$, which takes as input $(x,y)$ and returns $\text{EXT}(x,y)$, from which we keep only the first $k$ bits. (Note: A randomness extractor is a type of function that plays a central role in the theory of pseudo-randomness. All we need here is that it satisfies Equation (4).)

From the function $\text{EXT}_k$, we go back to the graph representation and we obtain the "prefix" bipartite graph $G_{n,k} = (L = \{0,1\}^n, R_k = \{0,1\}^k, E_k \subseteq L \times R_k)$, where, in $G_{n,k}$, we merge the right nodes of $G_n$ that have the same prefix of length $k$. The left degrees in the

prefix graph do not change. However, the right degrees may change and, as $k$ becomes smaller, the right degrees typically become larger due to merging.

The requirement is that, for every subset $B \subseteq L$ of size $|B| \geq 2^k$, for every $A \subseteq R_k$,

$$\left| \frac{|E_k(B,A)|}{|B| \times D} - \frac{|A|}{|R_k|} \right| \leq \epsilon, \tag{4}$$

where $E_k(B,A)$ is the set of edges between $B$ and $A$ in $G_{n,k}$. (Note: This means that $G_{n,k}$ is a $(k,\epsilon)$ randomness extractor.)

We also want to have the guarantee that each right node in $G_{n,t}$ has degree at least $\Delta$, where $\Delta$ and $t$ are parameters.

Accordingly, we have the following definition.

**Definition 2.** *A graph $G_n = (L, R, E \subseteq L \times R)$ as above is $(\epsilon, \Delta, t)$-balanced if the following requirements hold:*

1. *For every $k \in \{1, \ldots, n\}$, let $G_{n,k}$ be the graph corresponding to $\mathrm{EXT}_k$ described above. We require that, for every $k \in \{1, \ldots, n\}$, $G_{n,k}$ is a $(k, \epsilon)$ extractor, i.e., $G_{n,k}$ has the property in Equation (4).*
2. *In the graph $G_{n,t}$, every right node with non-zero degree has degree at least $\Delta$.*

In our application, we need balanced graphs in which the neighbors of a given node can be found effectively. As usual, we consider families of graphs $(G_n)_{n \geq 1}$ and we say that such a family is *computable* if there is an algorithm that takes as input $(x, y)$, views $x$ as a left node in $G_{|x|}$, views $y$ as the label of an edge outgoing from $x$ and outputs $z$, where $z$ is the right node where the edge $y$ lands in $G_{|x|}$.

The following lemma provides the balanced graphs that we need as explained in the proof overview in Section 2.

**Lemma 2.** *For every rational $\epsilon > 0$, there exist some constant $c$ and a computable family of graphs $(G_n)_{n \geq 1}$, where each $G_n = (L = \{0,1\}^n, R = \{0,1\}^n, E \subseteq L \times R)$ is $(\epsilon, \Delta, t)$-balanced graph, with left degree $D = 2^d$ for $d = \lceil \log(2n/\epsilon^2) \rceil$, $\Delta = 2(1/\epsilon)^{3/2}D$ and $t = n - \log \log n - c$.*

The proof of Lemma 2 is by the standard probabilistic method and is presented in Section 4.3.

*4.2. From Balanced Graphs to Increasing the Kolmogorov Complexity in the List-Approximation Sense*

The following lemma shows a generic transformation of a balanced graph into a function that takes as input $x$ and produces a list so that most of its elements have a complexity larger than $C(x)$.

**Lemma 3.** *Let us suppose that, for every $\delta > 0$, there are $t = t(n)$ and a computable family of graphs $(G_n)_{n \geq 1}$, where each $G_n = (L_n = \{0,1\}^n, R_n = \{0,1\}^n, E_n \subseteq L_n \times R_n)$ is $(\delta^2, \Delta, t)$-balanced graph, with $\Delta = 2(1/\delta^3) \cdot D$, where $D$ is the left degree.*

*Then, there exists a computable function $f$ that takes as input a string $x$ and a rational number $\delta > 0$ and returns a list containing strings of length $|x|$; additionally, the following are true:*

1. *The size of the list is $O((1/\delta)^3 D^2)$;*
2. *If $C(x) \leq t$, then $(1 - O(\delta))$ of the elements in the list have a complexity larger than $C(x)$.*

*(The constants hidden in $O(\cdot)$ do not depend on $\delta$.)*

**Proof.** The following arguments are valid if $\delta$ is smaller than some small positive constant. We assume that $\delta$ satisfies this condition and also that it is a power of $1/2$. This can be performed because scaling down $\delta$ by a constant factor only changes the constants in the

$O(\cdot)$ in the statement. Let $\epsilon = \delta^2$. We explain how to compute the list $f(x)$, with the property stipulated in the theorem's statement.

We take $G_n$ to be the $(\epsilon, \Delta, t)$-balanced graph with left nodes of length $n$ promised by the hypothesis. Let $G_{n,t}$ be the "prefix" graph obtained from $G_n$ by cutting the last $n - t$ bits in the labels of right nodes (thus preserving the prefix of length $t$ in the labels).

The list $f(x)$ is computed in two steps:

1. First, we view $x$ as a left node in $G_{n,t}$ and take $N(x)$, the multiset of all neighbors of $x$ in $G_{n,t}$.

2. Secondly, for each $p$ in $N(x)$, we take $A_p$ to be a set of $\Delta$ neighbors of $p$ in $G_{n,t}$ (e.g., the first $\Delta$ ones in some canonical order). We set $f(x) = \bigcup_{p \in N(x)} A_p$ (if $p$ appears $n_p$ times in $N(x)$, we also take $A_p$ in the union $n_p$ times; note that $f(x)$ is a multiset).

Note that all the elements in the list have length $n$ and the size of the list is $|f(x)| = \Delta \cdot D = 2(1/\delta)^3 D^2$.

Let $x$ be a binary string of length $n$, with complexity $C(x) = k$. We assume that $k \leq t$. The rest of the proof is dedicated to showing that the list $f(x)$ satisfies the second item in the statement. Let

$$B_{n,k} = \{x' \in \{0,1\}^n \mid C(x') \leq k\},$$

and let $S_{n,k} = \lfloor \log |B_{n,k}| \rfloor$. Thus, $2^{S_{n,k}} \leq |B_{n,k}| < 2^{S_{n,k}+1}$. Later, we use the fact that

$$S_{n,k} \leq k \leq t. \tag{5}$$

We consider the graph $G_{n,S_{n,k}}$, which is obtained, as explained above, from $G_n$ by taking the prefixes of the right nodes of length $S_{n,k}$. To simplify notation, we use $G$ instead of $G_{n,S_{n,k}}$. The set of left nodes in $G$ is $L = \{0,1\}^n$ and the set of right nodes in $G$ is $R = \{0,1\}^m$, for $m = S_{n,k}$.

We view $B_{n,k}$ as a subset of the left nodes in $G$. Let us introduce some helpful terminology. In the following, all the graph concepts (left node, right node, edge and neighbor) refer to the graph $G$. We say that a right node $z$ in $G$ is $(1/\epsilon)$-light if it has at most $(1/\epsilon) \cdot \frac{|B_{n,k}| \cdot D}{|R|}$ neighbors in $B_{n,k}$. A node that is not $(1/\epsilon)$-light is said to be $(1/\epsilon)$-heavy. Note that

$$(1/\epsilon) \cdot \frac{|B_{n,k}| \cdot D}{|R|} \leq (1/\epsilon) \frac{2^{S_{n,k}+1} \cdot D}{2^{S_{n,k}}} = \delta\Delta,$$

thus, a $(1/\epsilon)$-light node has at most $\delta\Delta$ neighbors in $B_{n,k}$.

We also say that a left node in $B_{n,k}$ is $\delta$-BAD with respect to $B_{n,k}$ if at least a $\delta$ fraction of the $D$ edges outgoing from it lands in the right neighbors that are $(1/\epsilon)$-heavy. Let $\delta$-BAD$(B_{n,k})$ be the set of nodes that are $\delta$-BAD with respect to $B_{n,k}$.

We show the following claim.

**Claim 2.** *At most a $2\delta$ fraction of the nodes in $B_{n,k}$ is $\delta$-BAD with respect to $B_{n,k}$.*

*(In other words, for every $x'$ in $B_{n,k}$ except at most a $2\delta$ fraction, at least a $(1 - \delta)$ fraction of the edges going out from $x'$ in $G$ lands in the right nodes that have at most $\Delta'$ neighbors with complexity at most $k$).*

We defer for later the proof of Claim 2 and continue the proof of the theorem.
For any positive integer $k$, let

$$B_k = \{x' \mid C(x') \leq k \text{ and } k \leq t(|x'|)\}.$$

Let $I_k = \{n \mid k \leq t(n)\}$. Note that $|B_k| = \sum_{n \in I_k} |B_{n,k}|$. Let $x' \in B_k$ and let $n' = |x'|$. We say that $x'$ is $\delta$-BAD with respect to $B_k$ if, in $G_{n'}$, $x'$ is $\delta$-BAD with respect to $B_{n',k}$. We

denote by $\delta$-BAD($B_k$) the set of nodes that are $\delta$-BAD with respect to $B_k$. We upper bound the size of $\delta$-BAD($B_k$) as follows:

$$
\begin{aligned}
|\delta\text{-BAD}(B_k)| \ & = \sum_{n' \in I_k} |\delta\text{-BAD}(B_{n',k})| \\
& \leq \sum_{n' \in I_k} 2\delta \cdot |B_{n',k}| \quad \text{(by Claim 2)} \\
& = 2\delta \sum_{n \in I_k} |B_{n',k}| \\
& = 2\delta |B_k| \\
& \leq 2\delta \cdot 2^{k+1}.
\end{aligned}
$$

Note that the set $\delta$-BAD($B_k$) can be enumerated given $k$ and $\delta$. Therefore, a node $x'$ that is $\delta$-BAD with respect to $B_k$ can be described by $k$, $\delta$ and its ordinal number in the enumeration of the set $\delta$-BAD($B_k$). We write the ordinal number on exactly $k + 2 - \log(1/\delta)$ bits and $\delta$ in a self-delimited way on $2 \log \log(1/\delta)$ bits (recall that $1/\delta$ is a power of 2), so that $k$ can be inferred from the ordinal number and $\delta$. It follows that, if $x'$ is $\delta$-BAD with respect to $B_k$, then, provided $1/\delta$ is sufficiently large,

$$
C(x') \leq k + 2 - \log(1/\delta) + 2 \log \log(1/\delta) + O(1) < k. \tag{6}
$$

Now, we recall our string $x \in \{0,1\}^n$, which has complexity $C(x) = k$. The inequality (6) implies that $x$ cannot be $\delta$-BAD with respect to $B_k$, which means that $(1 - \delta)$ of the edges going out from $x$ land in neighbors in $G$ having at most $\delta\Delta$ neighbors in $B_k$. The same is true if we replace $G$ by $G_{n,t}$, because, by the inequality (5), the right nodes in $G$ are prefixes of the right nodes in $G_{n,t}$.

Now, let us suppose that we pick at random a neighbor $p$ of $x$ in $G_{n,t}$ and then find a set $A_p$ of $\Delta$ neighbors of $p$ in $G_{n,t}$. Then, with probability $1 - \delta$, only a fraction of $\delta$ of the elements of $A_p$ can be in $B_k$. Let us recall that we have defined the list $f(x)$ to be

$$
f(x) = \bigcup_{p \text{ neighbor of } x \text{ in } G_{n,t}} A_p.
$$

It follows that at least a $(1 - \delta)^2 > (1 - 2\delta)$ fraction of the elements in $f(x)$ has complexity larger than $C(x)$. This ends the proof. □

We now prove Claim 2.

**Proof of Claim 2.** Let $A$ be the set of right nodes that are $(1/\epsilon)$-heavy. Then,

$$
|A| \leq \epsilon |R|.
$$

Indeed, the number of edges between $B_{n,k}$ and $A$ is at least $|A| \cdot (1/\epsilon) \cdot \frac{|B_{n,k}| \cdot D}{|R|}$ (by the definition of $(1/\epsilon)$-heavy), but, at the same time, the total number of edges between $B_{n,k}$ and $R$ is $|B_{n,k}| \cdot D$ (because each left node has degree $D$).

Next, we show that

$$
|\delta\text{-BAD}(B_{n,k})| \leq 2\delta |B_{n,k}|. \tag{7}
$$

For this, note that $G$ is a $(S_{n,k}, \epsilon)$ randomness extractor and $B_{n,k}$ has size at least $2^{S_{n,k}}$. Therefore, by the property (4) of extractors,

$$
\frac{|E(B_{n,k}, A)|}{|B_{n,k}| \cdot D} \leq \frac{|A|}{|R|} + \epsilon \leq 2\epsilon.
$$

On the other hand, the number of edges linking $B_{n,k}$ and $A$ is at least the number of edges linking $\delta$-BAD($B_{n,k}$) and $A$; this number is at least $|\delta\text{-BAD}(B_{n,k})| \cdot \delta D$. Thus,

$$
|E(B_{n,k}, A)| \geq |\delta\text{-BAD}(B_{n,k})| \cdot \delta D.
$$

Combining the last two inequalities, we obtain

$$\frac{|\delta\text{-BAD}(B_{n,k})|}{|B_{n,k}|} \leq 2\epsilon \cdot \frac{1}{\delta} = 2\delta.$$

This ends the proofs of Claim 2, which is the last piece that we needed for the proof of Lemma 3. □

Theorem 1 is obtained by plugging, into the above lemma, the balanced graphs from Lemma 2 with parameter $\epsilon = \delta^2$.

*4.3. Construction of Balanced Graphs: Proof of Lemma 2*

We use the probabilistic method. We consider a random function $\text{EXT} : \{0,1\}^n \times \{0,1\}^d \to \{0,1\}^n$ for $d = \lceil \log(2n/\epsilon^2) \rceil$. We show the following two claims, which imply that a random function has the desired properties with positive probability. Since the properties can be checked effectively, we can find a graph by exhaustive search. We use the notation from Definition 2 and from the paragraph preceding it.

**Claim 3.** *For sufficiently large n, with probability $\geq 3/4$, it holds that, for every $k \in \{1, \ldots, n\}$, in the bipartite graph $G_{n,k} = \{L, R_k, E_k \subseteq L \times R_k\}$, every $B \subseteq L = \{0,1\}^n$ of size $|B| \geq 2^k$ and every $A \subseteq R_k = \{0,1\}^k$ satisfies*

$$\left| \frac{|E_k(B,A)|}{|B| \times D} - \frac{|A|}{|R_k|} \right| \leq \epsilon. \tag{8}$$

**Claim 4.** *For some constant c and every sufficiently large positive integer n, with probability $\geq 3/4$, every right node in the graph $G_{n,n-\log\log n - c}$ has degree at least $\Delta$.*

**Proof of Claim 3.** First, we fix $k \in \{1, \ldots, n\}$ and let $K = 2^k$ and $N = 2^n$. Let us consider $B \subseteq \{0,1\}^n$ of size $|B| \geq K$ and $A \subseteq R_k$. For a fixed $x \in B$ and $y \in \{0,1\}^d$, the probability that $\text{EXT}_k(x, y)$ is in $A$ is $|A|/|R_k|$. By the Chernoff bounds,

$$\text{Prob}\left[ \left| \frac{|E_k(B,A)|}{|B| \times D} - \frac{|A|}{|R_k|} \right| > \epsilon \right] \leq 2^{-\Omega(K \cdot D \cdot \epsilon^2)}.$$

The probability that relation (8) fails for a fixed $k$, some $B \subseteq \{0,1\}^k$ of size $|B| \geq K$ and some $A \subseteq R_k$ is bounded by $2^K \cdot \binom{N}{K} \cdot 2^{-\Omega(K \cdot D \cdot \epsilon^2)}$, because $A$ can be chosen in $2^K$ ways; further, we can consider that $B$ has size exactly $K$ and that there are $\binom{N}{K}$ possible choices of such $B$'s. Since $D \geq 2n/\epsilon^2$, the above probability is much less than $(1/4)2^{-k}$. Therefore, the probability that relation (8) fails for some $k \in \{1, \ldots, n\}$, some $B$ and some $A$ is less than $1/4$. □

**Proof of Claim 4.** We use a "coupon collector" argument. We consider the graph $G_{n,n-\log\log n - c}$ for some constant $c$ to be fixed later. This graph is obtained from the above function EXT as explained in Definition 2. The graph $G_{n,n-\log\log n - c}$ is a bipartite graph with left side $L = \{0,1\}^n$, right side $R' = \{0,1\}^{n-\log\log n - c}$ and each left node has degree $D = 2^d$. We show that, with probability $\geq 3/4$, every right node in $G_{n,n-\log\log n - c}$ has degree at least $\Delta$. The random process consists of drawing, for each $x \in L$ and edge $y \in \{0,1\}^d$, a random element from $R'$. Thus, we draw at random $ND$ times, with replacement, from a set with $|R'|$ "coupons". Newman and Shepp [7] have shown that, to obtain at least $h$ times each coupon from a set of $p$ coupons, the expected number of draws is $p \log p + (h-1)p \log \log p + o(p)$. By Markov's inequality, if the number of draws is 4 times the expected value, we collect each coupon $p$ times with probability $3/4$. In our case, we have $p = 2^{n-\log\log n - c}$ and $h = \Delta$; it can be checked readily that, for an appropriate choice of the constant $c$, $4(p \log p + (h-1)p \log \log p + o(p)) < ND$, provided $n$ is large enough. □

**Funding:** The author has been supported in part by the National Science Foundation through grant CCF 1811729.

**Institutional Review Board Statement:** Not applicable.

**Informed Consent Statement:** Not applicable.

**Data Availability Statement:** Not applicable.

**Acknowledgments:** The author is grateful to Bruno Bauwens for his insightful observations and to Nikolay Vereshchagin for pointing out an error in an earlier version. The author thanks the anonymous referees for their useful suggestions.

**Conflicts of Interest:** The author declares no conflict of interest..

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
