# Peer review of "List Approximation for Increasing Kolmogorov Complexity"

_axioms, doi:10.3390/axioms10040334_

Round 1

Reviewer 1 Report

The paper was well-written and interesting. The main results are a trio of theorems showing that there is an effective way to transform most strings x into a small list of string of the same length most of which have higher Kolmogorov complexity (for various meanings of small and most).

I have two major comments. The first is that in the proof of Theorem 2, as far as I can see the number of random bits is O(log|x|/epsilon), which would make the list of size polynomial in |x|/epsilon, in which case this is strictly weaker than Theorem 1.

Second, in all three Theorems, the complexity of the elements in the list f(x) might only be C(x) + 1. This seems like not much of an improvement, especially as the lengths of the strings might be very long. In some sense, this is really no improvement in the randomness as 1/|x| is very small. If one picks a single element out of a list, then it is possible to get the randomness to go up even divided by the length of the string. My guess is that it is not possible to do this for a large proportion of the list? It might be good to include a short discussion of what you think one gets out of the theorems, and what the naive limitations/upper bounds are.

Minor comments / typos:

  1. Theorem 1 and 2 have poly(1/ffi), should be 1/epsilon I think.
  2. Section 1.2, end of first paragraph: "this has can be viewed"
  3. Just before Section 2: "namely Theorems 5 and 6"
  4. Proof of Theorem 3, just before (3): "The choice of c ensures" sounds like (3) is implied by (2), maybe instead say "Also choose c so that".
  5. Proof of Theorem 3: Technically yz is supposed to have length n, but this yz has length less than n. So maybe choose z to be c' random bits and then a bunch of 0's.
  6. Section 4.1, bullet 2: It wasn't immediately clear that there is no repitition of labels. Maybe say "labeled by the binary strings"
  7. I think we never actually need to know what it means for a function to be a randomness extractor -- maybe say this, so that the reader doesn't need to look up the definition if they don't remember it. Also, lower down in 4.1, maybe say that (4) is the definition of Gnk being an extractor? Otherwise you have introduced what it means for a function to be an extractor, but not (strictly speaking) a graph.
  8. Lemma 2: The family of graphs is computable, each individual graph is already computable since it is finite.
  9. Lemma 3: In Delta = ..., there is an extra \cdot.
  10. In the second paragraph, I think it should be prefix of length "t", not "t - a"
  11. After bullets 1 and 2, there is a "2" missing from Delta * D = (1/delta)^3 D^2
  12. B_{n,k} instead of B_{n.k} just below that. (Comma instead of period)
  13. Taking an x with C(k) = k <= t in the first paragraph seems too early; I think that line should be moved down to the paragraph "The rest of the proof..."
  14. It seems to me that in the last line of the proof of Lemma 3, you get (1-delta)^2 rather than (1-2delta). Of course it will all still work by taking delta a little smaller.
  15. In the proof of Claim 7, I took me a little to realise why the number of edges linking B_{n,k} and A is at least the number linking delta-Bad and A. Maybe just write "linking delta-Bad(B) \subseteq B_{n,k} and A".
  16. Same line, there is a ")" missing at the end.
  17. Proof of Claim 9, there is an |R|' which should be |R'|

Author Response

I am truly grateful to the two reviewers for their careful reading and useful suggestions. I have followed all their indications, with one exception, namely the discussion of  whether increasing the complexity by more than 1 bit is possible, which reviewer 1 has recommended. Clearly, the procedure in the paper can be iterated and larger increases can be obtained, but discussing parameters would be tedious and I feel that this would be outside the scope of the paper, which simply studies the circumstances in which *any* increase of complexity can be effectively obtained.

I have dropped  Th 2 from the initial submission, because as reviewer 1 has observed that result is fully overridden by Th 1 (there was a mistake in the statement, the list is not linear, but polynomial, and this is obtained already in Th 1).   So now there are two main results: Th 1 (which is the same Th 1 as in the initial submission) and Th 2 (which was labeled Th 3 in the initial submission).

Reviewer 2 Report

The suggestions are in the PDF.

Author Response

(The authors gave the same response as above.)

Round 2

Reviewer 1 Report

The author has made the changes requested.

Reviewer 2 Report

It seems that my comments were sufficiently addressed. Thank you!